ecology, palaeontology, environmental science

turbid reefs, coral cover, reef accretion, sea level, Great Barrier Reef

**Author for correspondence:**
Kyle M. Morgan
e-mail: kmorgan@ntu.edu.sg

# Projections of coral cover and habitat change on turbid reefs under future sea-level rise

Kyle M. Morgan[1,2], Chris T. Perry[2], Rudy Arthur[3], Hywel T. P. Williams[3] and Scott G. Smithers[4]

[1]Asian School of the Environment, Nanyang Technological University, Singapore, Singapore
[2]School of Geography, College of Life and Environmental Sciences, and [3]Computer Science, College of Engineering, Mathematics and Physical Sciences, University of Exeter, Exeter, UK
[4]Land and Water Science, College of Science and Engineering, James Cook University, Townsville, Australia

(iD) KMM, 0000-0002-3412-703X; CTP, 0000-0001-9398-2418; SGS, 0000-0002-4420-1897

Global sea-level rise (SLR) is projected to increase water depths above coral reefs. Although the impacts of climate disturbance events on coral cover and three-dimensional complexity are well documented, knowledge of how higher sea levels will influence future reef habitat extent and bioconstruction is limited. Here, we use 31 reef cores, coupled with detailed benthic ecological data, from turbid reefs on the central Great Barrier Reef, Australia, to model broad-scale changes in reef habitat following adjustments to reef geomorphology under different SLR scenarios. Model outputs show that modest increases in relative water depth above reefs (Representative Concentration Pathway (RCP) 4.5) over the next 100 years will increase the spatial extent of habitats with low coral cover and generic diversity. More severe SLR (RCP8.5) will completely submerge reef flats and move reef slope coral communities below the euphotic depth, despite the high vertical accretion rates that characterize these reefs. Our findings suggest adverse future trajectories associated with high emission climate scenarios which could threaten turbid reefs globally and their capacity to act as coral refugia from climate change.

## 1. Introduction

Coral reefs have undergone major declines in coral cover and biodiversity over recent decades [1,2]. Local pressures associated with resource over-exploitation and declining water quality have driven ecological changes on reefs [3]. However, these stressors are compounded by increasingly frequent and widespread thermal coral bleaching events that are the result of global climate change [4]. Such large-scale disturbances have significantly reduced reef-building coral taxa at many locations [5]. Potential impacts include lower rates of reef carbonate production [6], diminished vertical reef growth capacity [7] and a progressive flattening of three-dimensional (3D) reef structure as coral populations have transitioned to lower profile stress-tolerant communities [8]. The negative consequences of these changes for marine biodiversity and the geomorphological functions of coral reefs are of growing concern [9,10]. While we have a good understanding of the magnitude and drivers of ecological decline on reefs, a fundamental question remains: how will the spatial extent of reef habitat, that influence the physical functional role of reefs, change as global sea levels rise? This is not an easy question to address. Firstly, because for many reefs, the past is a very poor proxy for the present (and the future) as species compositions on reefs are often highly modified [9,11]. Secondly, there are very few sites where sufficient spatial knowledge of past reef accretion rates and associated changes in historical coral communities are available to inform models of future reef growth.

Here, we address this knowledge gap using an unprecedented dataset of reef cores (31 cores from seven proximal reefs) and reef benthic cover data collected from nearshore turbid reefs on the central Great Barrier Reef (GBR), Australia, to project future changes in reef morphology and habitat extent under different sea-level rise (SLR) scenarios. This is possible because recent work on these reefs has shown: (i) the long-term (centennial to millennial timescale) persistence of coral communities under high terrestrial sedimentation [12,13]; (ii) a very strong correlation between past and present coral assemblages [14,15]; and (iii) a similarly strong correlation between depth-controlled habitats and their vertical accretion rates [16]. The relationship between habitat type and the relative water level above reefs results from the rapidly changing light, wave and sedimentary conditions within coastal areas of the GBR [17] that may also provide turbid corals with a greater resistance to thermal bleaching [18]. The continued growth of reef-building corals from past through to present-day reefscapes in the absence of major disturbance (e.g. coral bleaching), and the consistent nature of reef accretion rates associated with depth-constrained habitat zones, provides a unique opportunity to model future reef behaviour and change.

Prior knowledge of the spatio-temporal dynamics of turbid reefs is applied here to construct a depth/habitat-controlled predictive reef growth model. The model allows for reef growth to be simulated, beginning from the present-day morphology, through time under different scenarios of SLR. Model outputs are used to quantify changes in habitat extent following adjustments of the underlying reef morphology as it grows vertically. Based on detailed spatial data on the living coral communities, we can then infer changes in key ecological metrics (e.g. coral cover, coral composition, reef rugosity). Reverse-time model runs also allow us to validate outputs against observed age/depth histories from reef cores, giving new broad-scale insights into the early onset of turbid reefs and their coral communities. Our results show the long-term ecological dynamics of turbid reefs and provide robust projections of future reef trajectories. This is important because turbid reefs have been identified as potential climate change refugia for corals [18–20], and as projections suggest increasingly frequent thermal disturbances on tropical reefs [21,22], it is critical to evaluate how habitats may change in response to reef geomorphological development and to sea-level changes.

## 2. Material and methods

### (a) Site description

The Paluma Shoals reef complex (PSRC) is located less than 3 km off the Queensland coast within Halifax Bay (19.1145° S, 146.5497° W) on the central GBR, Australia (electronic supplementary material, figure S1). PSRC is a series of linear shore-perpendicular coral reefs comprising: (i) shore-attached reefs (Paluma Shoals (PS)) located within shallower water (less than 4 m depth) and adjoining intertidal sand flats at the coast. The reefs have recently reached sea level and become emergent under low tidal conditions (+0.5 m above lowest astronomical tide; LAT); and (ii) shore-detached reefs (Offshore Paluma Shoals (OPS) A–D) which remain submerged in deeper water further offshore (3–6 m LAT). PSRC experiences high turbidity (up to 385 mg l$^{-1}$) and low-light conditions because of the wave and tidal resuspension (tidal range: 3.6 m) of fine seafloor material that was reworked

onshore during the last post-glacial transgression [12]. Light attenuation (97% reduction by 4 m LAT) drives rapid shifts in coral assemblages over vertically compressed depth ranges and restricts coral growth to approximately 4 m below LAT [15].

### (b) Field data collection

To inform our model, we collected field data on the spatial distribution of coral communities, the morphology of present-day reefs and their Holocene growth histories [14–16]. The broad extent of reefs was established from Landsat imagery and acoustic seafloor surveys of the area. A single-beam echo sounder (Ceeducer Ceestar 200 kHz) coupled with a real-time kinematic global positioning system (RTK-GPS) was used to obtain a high-precision digital elevation model (DEM) of the seafloor and reef morphology. Benthic ecological surveys were conducted (July 2013, 2014 and 2016) across the seafloor using a boat-towed drop-down video system (SeaViewer with Sea-Track™ GPS overlay). The camera was towed approximately 1 m above the seafloor from which digitally geo-referenced still frames ($n = 4420$) were automatically extracted from the video at 8 s intervals. Each frame was depth-calibrated using the DEM and a digital 9-point grid overlay was added to frames to calculate benthic community composition (%) and make a qualitative estimate (scaled 1–5) of reef rugosity ($R$) [15]. The depth ranges of dominant coral genera and key habitat types were then established (H1: *Goniastrea* reef flat (>LAT); H2: rubble and encrusting coral (0–0.5 m); H3: *Montipora* and *Acropora* framework (0.5–1.5 m); H4: massive *Porites* and sand (1.5–2 m); H5: *Turbinaria* carpets (2–3 m); H6: rubble and sediment-tolerant coral (3–4 m); H7: sand/mud (greater than 4 m)).

To constrain rates of vertical accretion in our model, we used data from previously published core records (16 reef cores from submerged OPS–OPSD and 15 cores from the sea-level attained PS) collected using percussion coring techniques [16,23,24]. This method allows for 100% recovery of undisturbed coral framework and sediment matrix. Each core terminated in pre-reefal substrate at their base. *In situ* coral material was selected for radiocarbon dating (142 radiometric dates from 31 cores) to establish the age and accretionary history of reefs. Rates of vertical reef accretion (mm yr$^{-1}$) were calculated between consecutive dated corals in the core, and then pooled to determine average rates of vertical reef accretion for each 0.5 m depth interval relative to the LAT (electronic supplementary material, table S1). Palaeoecological analysis of coral material shows a direct comparison between living and historical coral communities within similar depth ranges [14], and that observed average rates of vertical reef accretion closely align with these shifts in coral assemblages as the reefs shallow [16].

### (c) Model description

The DEM of seafloor and reef morphology (total area: 15 km$^2$; $10 \times 10$ m grid cells) was used to model past and future reef growth (±500 years) and to quantify habitat extent. We first assigned each grid cell a label corresponding to the absence or presence of reef ($P \in \{0,1\}$). A small number of grid cells exceeded 0.5 m LAT (i.e. above sea level) and below the 4 m LAT depth contour (limit of coral growth). For simplicity, we forced all grid cell heights into −4.0 to +0.5 m depth/elevation. The thickness of Holocene reef (i.e. vertical distance from the reef surface to the underlying sediment) was determined as the termination point at which reverse-time extrapolations should cease. This could not be precisely constrained across the entire complex from cores, but was predicted from measured depths of inter-reefal seafloor areas. We estimate seafloor depth using distance weighted $k$-nearest neighbour interpolation. At $T = 0$, for every reef site, $(X,Y)$, we identify the $k$ closest non-reef sites, $c$, and

their distances from the target site, $D(c)$. We then estimate the seafloor depth at $(X,Y)$ as

$$s(X,Y) = \frac{\sum_c^k w_c \, d(X_c,Y_c,0)}{\sum_c^k w_c}, \tag{2.1}$$

where $w_c = 1/D_C$. This approach makes nearby sites more important and a larger $k$ will lead to smoother interpolation. In our backwards extrapolation, we use a range of $k$ values: (1, 4, 10, 20, 50, 100, 250) to account for different possible seafloors and give our final error estimate based on the range of values produced across all extrapolations.

## (d) Estimating vertical and lateral reef growth

To calculate vertical reef growth and estimated error $\Delta$Rate, we used depth-variable accretion rates at 0.5 m intervals derived from cores (ranging from $1.4 \pm 1.1$ to $6.9 \pm 9.4$ mm yr$^{-1}$) (electronic supplementary material, table S1). We use the symbol $d(X,Y,T)$ for depth in year $T$. $T = 0$ corresponds to present day (2016) and thus $d(X,Y,0) = Z$. We then extrapolate backwards (equation (2.2)) assuming a static sea level as suggested by Late Holocene sea-level data for the region [25], or forwards (equation (2.3)), by subtracting or adding a growth term $g(d(X,Y,T))$. This term uses established annual reef accretion rates $g(d)$ and the depth $d(X,Y,T)$ of each grid cell containing coral at year $T$. We assume that if water depth over a cell is greater than 0.5 m elevation (emergent) or greater than 4.0 m depth (below the euphotic depth), corals will not grow: $g(d) = 0$ for $d > 4$ m or $d < 0.5$ m:

$$d(X,Y,T+1) = d(X,Y,T) + g(d(X,Y,T)) \tag{2.2}$$

and

$$d(X,Y,T-1) = d(X,Y,T) - g(d(X,Y,T)). \tag{2.3}$$

To account for uncertainty in reef growth rates, we use a bootstrap procedure. First, we assume the amount of vertical accretion at every grid cell is independent of every other. For each bootstrap sample, we choose the accretion rate at depth $d$ from a Gaussian distribution with mean and standard deviation given by the appropriate row (electronic supplementary material, table S1). We then run the whole extrapolation $B = 100$ times to produce a range of different outputs and calculate the average and the 95% confidence interval of our measurements.

To model lateral reef growth, we use a lateral expansion rate ($L = 0.25$ m yr$^{-1}$) calculated from the average age difference between consecutive reef cores at the same horizontal distance on a transect, and the horizontal distance between these cores [23]. We assume uniform expansion with depth, and model lateral reef expansion as follows: $p = L/s$, where $s$ is the side length of a grid cell, then $p$ is the proportion by which the reef grid cell expands each year into a neighbouring non-reef grid cell (independent of adjacent cell depth). This is applied to all grid cells which do not yet contain reef but have neighbouring cells that do contain reef. For each of these (where $R(T) = 0$ initially), we accumulate

$$R(T+1) = R(T) + p(\delta(1,R(X+1,Y)) + \delta(1,R(X-1,Y)) + \delta(1,R(X,Y+1)) + \delta(1,R(X,Y-1))), \tag{2.4}$$

where $\delta$ is the Kronecker delta: $\delta(i,j) = 1$ if $i = j$; otherwise, it is zero. This means we add a fraction $p$, from neighbouring reef cells to the tally for its neighbouring non-reef cells. When $R \geq 1$, the reef has covered the entire cell, and we stop the accumulation and allow the new reef cell to grow vertically at the designated depth-assigned accretion rate (i.e. vertical accretion only begins once the cell is filled), as well as contribute to reef expansion into neighbouring cells.

## (e) Calculating sea-level change and habitat extent

We then incorporated rates of SLR ($r$) projected by Intergovernmental Panel on Climate Change Representative Concentration

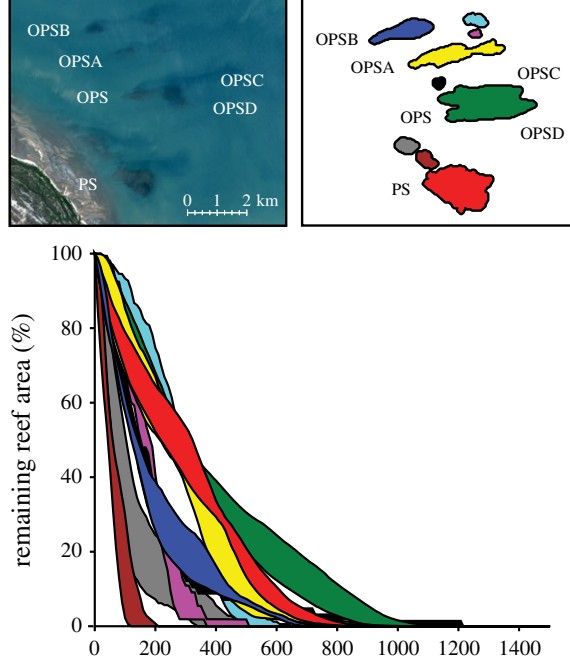

**Figure 1.** Per cent (%) of remaining reef area as reef morphology is modelled backwards from present day to reef initiation (i.e. start-up). Model simulations assume no sea-level change and constant (but depth-variable) reef accretion rates. Lines represent the min/max estimated change from a 95% confidence window using 100 bootstrap samples calculated for different seafloor extrapolations (see Material and methods). Colours indicate the reefs that form the PSRC (PS, OPS A–D). (Online version in colour.)

Pathways (RCP) 4.5 (5.5 mm yr$^{-1}$) and 8.5 (7.5 mm yr$^{-1}$) climate change scenarios [21]. We replace equation (2.2) with a modified form, equation (2.5), in which the value of $r$ is subtracted from the heights of all grid cells each year (after accounting for vertical reef growth):

$$d(X,Y,T+1) = d(X,Y,T) + g(d(X,Y,T)) - r. \tag{2.5}$$

Reef habitat extent was interpolated from modelled seafloor and reef morphology (forwards and backwards) at 100-year time-slices. Habitat maps were generated for each time period by allocating each grid cell a specific habitat type (H1–H7) based on the elevation/depth of that cell relative to LAT (electronic supplementary material, table S2), and established habitat depth ranges observed in both living and palaeo-coral communities. Total habitat extent was tallied across the reef surface and the relative cover of habitats through time was determined by calculating the number of grid cells of each habitat compared to all reef grid cells. Estimates of coral cover, coral composition and reef rugosity associated with each habitat type are inferred from ecological datasets of living coral communities at PSRC (electronic supplementary material, table S2).

## 3. Results

### (a) Model validation using reef cores

To test the model reliability, the model was first run backwards from present day until all reef cells were removed. By shrinking the reefs backwards, the model indicates that the main reef complexes started growing approximately 1200–700 years before present day (yrs BP) (figure 1). The modelled timings of reef initiation align with the reef-building phases discerned from the basal ages in cores at these sites (approx.

1400–700 yrs BP) [16]. There are several smaller reef structures evident in our bathymetry and ecological datasets. We do not have core data from these to test the model against, but assuming they have had similar growth histories to the more established main reefs, our model suggests these would have established somewhere between 600 and 200 yrs BP (figure 1). The agreement between model and core datasets suggests that core-derived accretion rates can be successfully used to predict past reef growth, giving confidence in using them to predict future morphological development over similar timescales and environmental conditions.

## (b) Modelled phases of past reef development and habitat extent

Core records suggest that by 500 yrs BP, reefs comprising sediment-tolerant coral genera (e.g. *Lobophyllia*, *Goniopora*, *Galaxea*) had initiated early framework accumulation that accreted slowly (2.4 mm yr$^{-1}$) for several hundred years [16]. By approximately 200 yrs BP, reefs transitioned to rapid rates of vertical reef growth ($5.1 \pm 4$ to $6.9 \pm 9.4$ mm yr$^{-1}$) as they reached water depths between 1.5 and 3.5 m, and fast-growing coral taxa (e.g. foliose *Montipora* and *Turbinaria*, branching *Acropora*) became more abundant [14,16]. Shore-attached reefs (e.g. PS) located within shallower water (less than 4 m depth) reached sea level less than 300 yrs BP, with reduced rates of vertical reef accretion (1.4 mm yr$^{-1}$) as coral communities shifted to lower abundance of slow-growing and exposure-tolerant taxa (e.g. *Galaxea*, *Goniastrea*) [16].

Our model shows highly consistent spatial patterns with the above core-derived time points as reefs are grown from initiation to the present day. The model shows that early reef growth (at 500 yrs BP) begins with isolated patch reefs (total reef area: 53 ha; figure 2). Habitat maps derived from the reef morphology indicate that early reefs (3–4 m depth) were dominated by rubble and sediment-tolerant (and potentially heterotrophic) coral habitat (H6) across approximately 50–60% of the reef complex surface between 500 and 300 yrs BP (figures 3 and 4). This habitat type is associated with coral communities defined by low coral cover ($11 \pm 9.9\%$) and low-profile framework (median rugosity ($R$) = 2) (electronic supplementary material, table S2). Model outputs then show that by 200 yrs BP, these patch reefs grew above the seafloor (+1.5 m) and expanded their spatial footprint (total reef area: 124 ha) (figure 2; electronic supplementary material, table S3). The changes in reef morphology led to the expansion of shallow-water *Montipora* and *Acropora* framework habitat (H3: 0.5–1.5 m depth), covering 24% of the total reef area, and deeper *Turbinaria* carpets habitat (H5: 2–3 m depth), covering 34% of total reef area at present day (figures 3 and 4). Both habitats, based on observations of living coral assemblages (electronic supplementary material, table S2), are associated with high coral cover and rugosity ($47 \pm 19\%$; median $R$ = 4 and $71.9 \pm 19.7\%$; median $R$ = 5, respectively).

## (c) Modelled projections of future change under a static sea level

The model was parametrized with static sea-level conditions and run forwards from the present day up to +500 years to determine how quickly reefs would fill their vertical accommodation space and form reef flats [26]. Models predict that as reefs accrete towards sea level, the spatial extent and relative cover of habitats will change (figures 3 and 4). The model shows declines in *Turbinaria* carpets habitat (H5: 2–3 m) to 10% of total reef area, and expansion of shallow-water *Montipora* and *Acropora* framework habitat (H3; 0.5–1.5 m depth) to 32% of total reef area by +300 years (figure 4). Coral communities associated with the *Montipora* and *Acropora* framework habitat are characterized by high coral cover ($47 \pm 19\%$; median $R$ = 4) and rapid vertical accretion (3.4 mm yr$^{-1}$) (electronic supplementary material, table S2). At these rates of reef growth, reef flat habitat (H1: >LAT) will form across 25% of reef surfaces (figures 3 and 4) between +300 and 500 years. Shifts in coral communities associated with these changes in habitat type suggest that coral cover on upper reef surfaces will decline from 47% at present day to 13% by +500 years, and transitions in benthic cover to exposure-tolerant massive corals (*Goniastrea*), encrusting corals (*Montipora*, *Galaxea*), rubble ($74 \pm 12.3\%$) and macroalgae ($2.5 \pm 3.9\%$) will occur (electronic supplementary material, table S2). These habitat reconfigurations will occur first on shore-attached reefs (e.g. PS at +100 years), followed by deeper shore-detached reefs (e.g. OPS–OPSD at +300–500 years).

## (d) Influence of sea-level rise on reef morphology and habitat extent (+100 years)

Sea level will not remain static in the future and, therefore, we also ran the reef growth model forwards (+100 years from present day) under average global rates of projected SLR (figure 5). Under the moderate emission scenario of RCP4.5 (5.5 mm yr$^{-1}$), a higher relative water level would mean that on shore-attached reefs (e.g. PS), habitat will shift from primarily *Goniastrea* reef flat habitat (H1) at present day to *Montipora* and *Acropora* framework habitat (H3) by +100 years (figure 5a,b). Based on observations of living coral communities, this would result in a transition to a high coral cover and reef complexity state (40–60%; $R$ = 3; figure 5c,d). Conversely, on shore-detached reefs (OPS–OPSD), higher relative water levels will submerge reefs (figure 5b), and the spatial extent of *Montipora* and *Acropora* framework habitat (H3) will decline to 5% by +100 years (figure 5a). Low coral cover and rugosity habitats (H6 and H7) will expand to 51% of the total reef area (figure 5c,d), as deeper reef slope coral communities (e.g. *Lobophyllia*, *Goniopora*, *Galaxea*) move below the euphotic depth.

Under the high emission scenario of RCP8.5 and higher rates of SLR (7.5 mm yr$^{-1}$), the models show exaggerated trends in habitat change (figure 5). These occur because the rates of projected SLR under RCP8.5 exceed even the highest rates of reef accretion that define these turbid reef complexes ($6.9 \pm 9.4$ mm yr$^{-1}$). The model predicts the loss of *Goniastrea* reef flat habitat (H1) at +100 years as present-day reef flats are submerged (figure 5a). At shore-attached (PS) reefs, these areas could become recolonized by *Montipora* and *Acropora* framework habitat (H3; figure 5a,b). However, higher relative water levels (0.25 m higher than RCP4.5 at +100 years) on shore-detached reefs (OPS–OPSD) are predicted to restrict *Montipora* and *Acropora* framework habitat (H3) to only nodal topographic areas. As a result, the model shows major losses of present-day shallow-water habitat and reductions in coral cover (to between 0 and 20%) and reef structural complexity ($R$ = 1–2) on upper reef surfaces by +100 years (figure 5c,d). Reef slope habitat (H6) will deepen to expand non-reef-building substrates (H7). Habitat maps also show the dominance

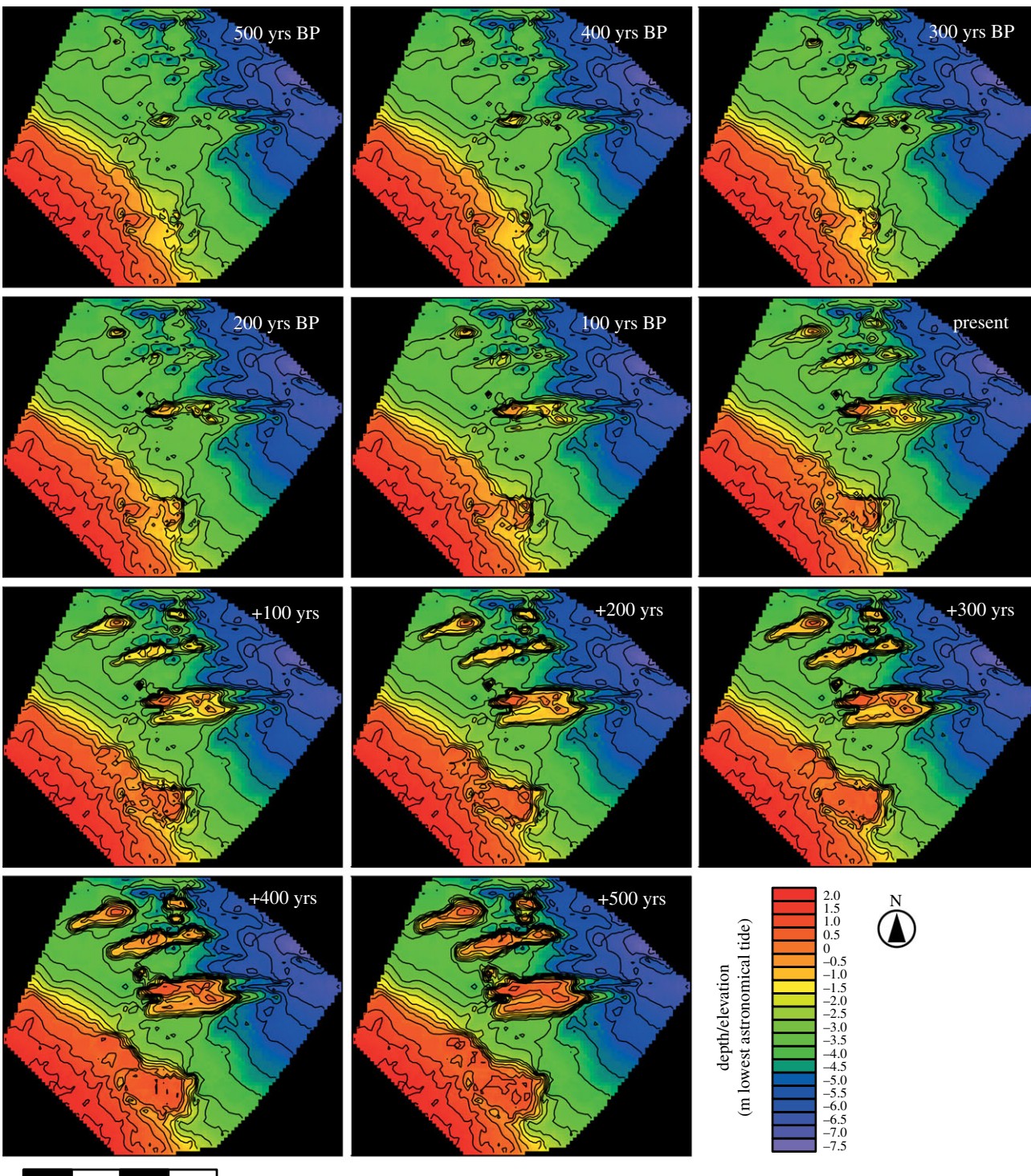

**Figure 2.** Geomorphic evolution (±500 years from present day) of PSRC, GBR, modelled under static sea level. Reef and seafloor morphology (metres relative to LAT) was modelled backwards and forwards from present-day bathymetry using depth-calibrated rates of vertical reef accretion derived from reef cores. (Online version in colour.)

of *Turbinaria* carpets (H3) as the main coral habitat on shore-detached reefs by +100 years (figure 5*a*), inhabiting deep and muddy waters (figure 5*b*), which comprise high coral cover (60–80%; R = 5), but low generic diversity of only very resilient coral taxa (electronic supplementary material, table S2).

## 4. Discussion

Turbid corals have been shown to exhibit: (i) a high-level resistance to thermal stress events [18,27], (ii) morphological

plasticity that allows corals to inhabit high sedimentation and low-light settings [28], and (iii) the ability to use hetero-trophy to sustain their energetic requirements under marginal conditions [29,30]. The resilience of turbid corals to fluctuating environmental conditions has led to sugges-tions that they may represent important refugia sites from climate change [19], and play a critical role in maintaining marine biodiversity in coastal areas. However, the capacity of turbid reefs to fulfil these functions will depend on how they evolve and respond to external forcing (e.g. SLR). Our model suggests that SLR could result in marked changes to

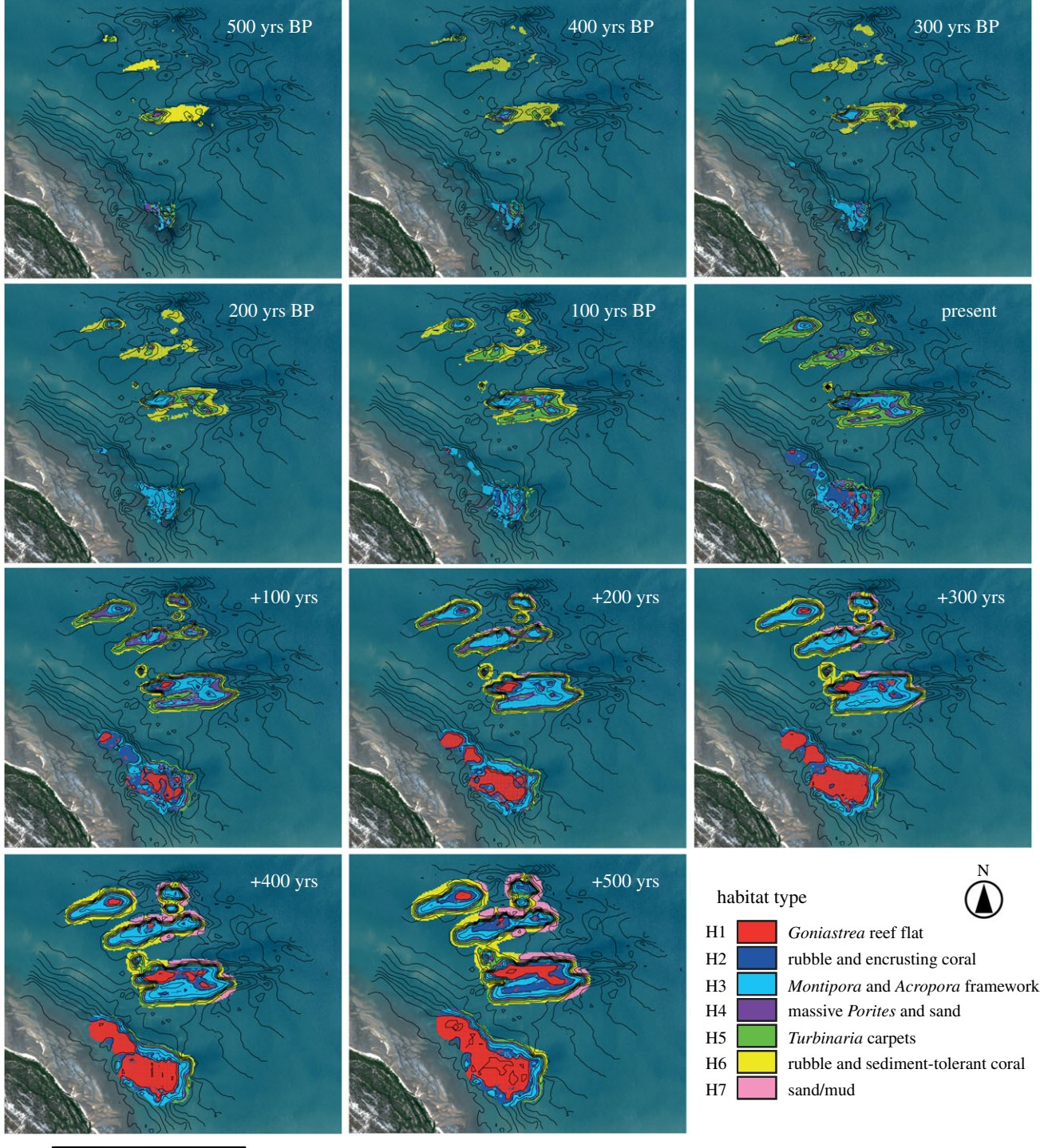

**Figure 3.** Spatial distribution of reef habitats throughout stages of reef geomorphic development (±500 years from present day). Habitat type (H1–H7) is determined by the relative water level above the reef surface as the reef (de)evolves, and associated coral communities are derived from observations of living coral depth ranges. Black lines denote the seafloor and reef bathymetry for each time interval. (Online version in colour.)

habitat extent and coral species composition. This is because the relative water level above reefs is a key driver of benthic habitat type within nearshore settings [15,31], as specific coral genera are adapted to cope with varying degrees of light, wave energy, aerial exposure and sedimentation. As these reefs build vertically, or are increasingly submerged by rising sea levels, these environmental conditions are dramatically altered by even relatively small water level changes.

Reef geomorphological evolution will therefore drive natural perturbations in coral cover, species composition and

reef structural complexity as growth of the underlying structure changes relative water depths above reefs [26]. Here, we show that shifts in habitat extent as reefs continue to develop over the next 100–500 years will directly impact ecosystem functioning by changing the abundance of coral taxa that: (i) construct different 3D configurations on reefs (and which impact upon biodiversity, fish biomass, etc.), (ii) modify rates of carbonate production and reef accretion, and (iii) influence the physical role of the reef structure in providing wave protection to adjacent shorelines [32]. Our findings suggest that PSRC is currently experiencing optimal growth (where coral cover averages

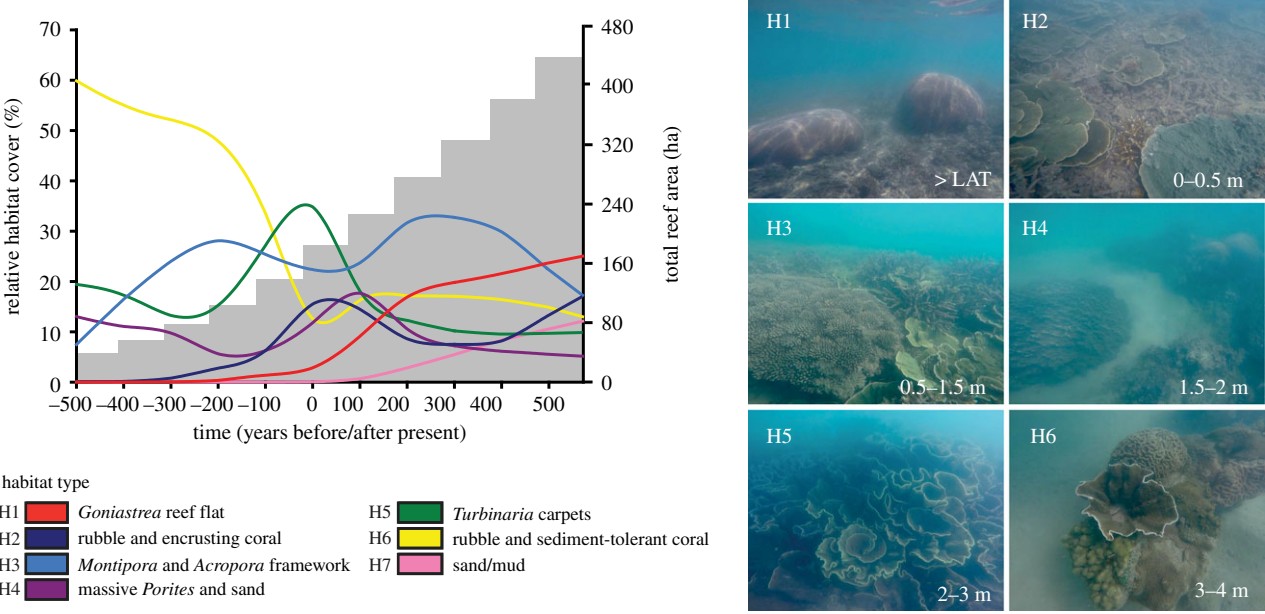

**Figure 4.** Relative habitat cover (%) in response to changes in reef morphology (±500 years from present day). Bars indicate the total planar reef area (hectares). Photographs (H1–H6, H7 not pictured) show each habitat type and their depth range (metres relative to LAT): (H1) *Goniastrea* reef flat; (H2) rubble and encrusting coral; (H3) *Montipora* and *Acropora* framework; (H4) massive *Porites* and sand; (H5) *Turbinaria* carpets; (H6) rubble and sediment-tolerant coral; (H7) sand/mud.

**Figure 5.** Projections of reef morphology (m relative to LAT) and the spatial distribution and relative cover (%) of reef habitats (H1–H7) across PSRC +100 years from present day. Models were run under a static sea level, and with sea-level rise based on rates projected by Representative Concentration Pathway (RCP) scenarios (RCP4.5 = 5.5 mm yr⁻¹; RCP8.5 = 7.5 mm yr⁻¹). Histograms show the total area (hectares) of each model category ((*a*) habitat type, (*b*) reef depth, (*c*) coral cover, (*d*) rugosity). Plots are separated into Paluma Shoals (shore-attached) and Offshore Paluma Shoals (shore-detached) reefs for each of the sea-level projections: static (total area: 56.3 ha), RCP4.5 (total area: 71.7 ha) and RCP 8.5 (total area: 72.3 ha). (Online version in colour.)

47%, but measures 100% over large areas) because of the high proportion of shallow-water habitat (0.5–1.5 m depth over 24% of total reef area). However, ongoing reef accretion with no major disturbance at these present rates (3.6 mm yr$^{-1}$) will fill vertical accommodation space within the next 500 years under a static sea level (figure 2) leading to a progressive 'turn-off' of carbonate production.

The timing and extent of future reef habitat reconfiguration is dependent on regional (and global) sea level. Past large-scale perturbations in benthic communities and reef accretion in response to relative sea-level change are widely documented. Sea-level fall on the GBR (1–1.3 m at approx. 2000 yrs BP) during the Late Holocene have previously caused the 'turn-off' of productive coral growth within shallow-water habitats, and a temporary hiatus of vertical reef growth as benthic communities shifted to rubble and macro-algal assemblages [33–35]. Conversely, emergent reef flats lowered by seismic subsidence of approximately 0.6 m have been successfully recolonized by coral framework [36]. These studies highlight the ecological dynamics of shallow reef communities when observed over geomorphological timescales, and provide justification to the types of relative ecosystem changes that we show here. Global SLR is projected to significantly increase water levels above reefs [7], and this will clearly influence the formation of reef flats and resultant coral productivity, but the extent to which external forcing drives ecological change ultimately depends on the rates of future sea-level change on the GBR.

Our models show that moderate rates of SLR under conditions where atmospheric $CO_2$ is stabilized to around 550 ppm by 2100 (RCP4.5 = 5.5 mm yr$^{-1}$) could have some positive outcomes by increasing vertical accommodation space and re-establishing productive coral growth on shore-attached (e.g. PS) reef flats. However, central to whether transitions back to high coral cover states are possible will depend on the health of living coral populations to provide a viable recruitment source [36], and the suitability of reef flat substrates for new coral settlement and growth. Although corals at PSRC are diverse and abundant [15], reef flat recolonization could be hindered by abundant turf algae that can competitively inhibit coral settlement on reefs [37], with experimental studies documenting higher recruitment success on substrates with low algal biomass [38]. Algal–coral interactions are further complicated on turbid reefs by the accumulation of turf-bound terrestrial sediment [39] that can also reduce coral recruitment [37].

Rapid SLR under a continued high emission scenario (RCP8.5 = 7.5 mm yr$^{-1}$) could raise water levels by at least 0.5 m by 2100 to amplify the above trends, with negative impacts on coral habitat availability (figure 5). This is concerning as recent climate projections suggest higher future rates of SLR are likely. Water depth increases above reefs create further vertical space for coral growth, but also significantly raises the euphotic depth (presently approx. 4 m LAT). Productive shallow-water coral communities may increase on shore-attached reefs (sea-level attained) as waters deepen. However, declines in coral diversity are predicted on shore-detached reefs (submerged) as light levels diminish and benthic communities become dominated by *Turbinaria* and *Porites* corals living in muddy waters (figure 5). Deeper water will also restrict productive coral habitat to small topographic high points (above 1.5 m depth), and push many coral populations (3–4 m depth) beyond their photic limits. Losses in coral cover and

reef structural complexity will occur, but these could be offset by new areas of coral growth as coastal areas are increasingly inundated and shoreline retreat promotes changes in nearshore bathymetry.

Although our study focuses on the PSRC (because of the extensive and detailed ecological and geological datasets available from these sites), other nearby reefs on the GBR show similar spatial distributions in habitat type and coral community structure [31]. Further, a recent global analysis using Modis-Aqua satellite imagery has suggested that approximately 12% of the world's coral reefs exist within a 'moderating turbidity' range, where turbidity is sufficient to mediate coral bleaching because of increased light attenuation [40]. Of the reefs located within the central and northern GBR ecoregion, 6.43% (562 km$^2$) were considered turbid [40]. This highlights the wider implications of our findings beyond PSRC as we expect to see similar shifts in habitat extent and coral cover within these other coastal areas as sea levels rise. At an individual reef-scale, the timing and magnitude of habitat reconfiguration will be determined by three interacting factors: (i) the rate of SLR; (ii) the rate of vertical reef accretion by coral communities; and (iii) the local turbidity regime. Reefs that inhabit more turbid water will respond quickly because coral depth ranges are likely to be very vertically compressed. Conversely, similar increases in relative water depth on clear-water reefs, where depth is less of a driver of community structure, could have less immediate ecological impacts.

## 5. Conclusion

Previous attempts to predict future reef growth trajectories have often been hindered by spatially limited core records, and distinct ecological divergence between the past and present reef communities that drive carbonate productivity [9,10]. The predictive model presented is unencumbered by these limitations. Our model generates assessments of reef functionality over timescales that far exceed even the most long-term benthic monitoring or geophysical studies. The extended timeframe, but high spatial resolution of outputs, from our model offers new quantitative insights into shifts in reef habitat type to relative changes in water depth not previously available. Results show that coral habitats are dynamic over geomorphic timescales and the distribution of reef habitat and coral communities is very dependent on the ongoing evolution of reef morphology. Global SLR will present a major threat to turbid coral populations as potential refugia from climate change by reducing shallow-water coral cover, reef structural complexity and coral diversity as available light on reefs diminishes. Understanding the geomorphological evolution of coastal settings is, therefore, critical for interpreting broad-scale future trends in coral distribution and habitat availability on reefs.

**Data accessibility.** All field datasets are available from the NERC datacentre: http://www.bgs.ac.uk/services/ngdc/accessions/index. html?simpleText=Great%20Barrier%20Reef#item76769. The model data that support the findings of this study are openly available at: https://github.com/rudyarthur/coral.

**Authors' contributions.** The study was conceived by K.M.M. and C.T.P. Model development was undertaken by R.A. and H.T.P.W. Fieldwork was conducted by K.M.M., C.T.P. and S.G.S. Data analysis was conducted by K.M.M. The manuscript was written by K.M.M. and C.T.P. All authors gave their approval for publication.

**Competing interests.** We declare we have no competing interests.

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
