## [Reviewer comments · Proceedings of the Royal Society B: Biological Sciences]

Review History

RSPB-2020-0541.R0 (Original submission)

Review form: Reviewer 1

Recommendation

Accept with minor revision (please list in comments)

Scientific importance: Is the manuscript an original and important contribution to its field?

Excellent

General interest: Is the paper of sufficient general interest?

Excellent

Quality of the paper: Is the overall quality of the paper suitable?

Excellent

Is the length of the paper justified?

Yes

Should the paper be seen by a specialist statistical reviewer?

No

Do you have any concerns about statistical analyses in this paper? If so, please specify them explicitly in your report.

No

It is a condition of publication that authors make their supporting data, code and materials available - either as supplementary material or hosted in an external repository. Please rate, if applicable, the supporting data on the following criteria.

Is it accessible?

Yes

Is it clear?

Yes

Is it adequate?

No

Do you have any ethical concerns with this paper?

No

Comments to the Author

The manuscript entitled 'Projections of coral cover and habitat loss on turbid reefs submerged by rising sea level' provides novel and useful insights into how these marginal reef types will respond to SLR in regards to their accretionary potential and habitat structure. The authors highlight the importance of understanding how these reefs will respond to SLR and the rates of change given that these reefs have been linked to increased resilience to sea surface temperature and may therefore be potential coral refugia hotspots. The manuscript is well written and the model development is (mostly) well explained. The modelling relies on one of the largest collection of reef cores available globally for a relatively small reef complex and together with detailed habitat maps of the present day reef - is based on a large and robust data set. As such, I recommend the paper for publication with minor changes (see below). Most of the required changes relate either to more clarification around terminology and methods, and the potential for new or replacement figures that better match the text descriptions.

Line 40: 'reductions in coral cover and diversity' I found it hard to actually 'see' the decline in coral cover in the manuscript. This was described in the text but not visually represented in an accessible manner - only relative changes in the habitat types. Further the term diversity here is misleading for two reasons. The authors are referring to habitat diversity rather than coral diversity (so could be mis-interpreted) but also in the figures - you can see that there is a change in the relative contribution of the 7 habitat types but all 7 are present most of the time - so no major changes in diversity at this level.

Line 42 to 44: I agree with this line - which is largely based on the finding that rates of SLR under RCP8.5 are above the fastest rates of accretion seen on these turbid reefs (6.9 mm/year). But this is quite fast generally right?? So could this statement be taken beyond turbid reefs or is this more significant on turbid reefs given the turbidity driven reduced light levels below 4 m depth?

Line 74: 'These marginal conditions...' This is a confusing sentence and sounds like the environment depend on corals. I know what you mean - but needs re-writing.

Line 82: 'reef configuration' - would 'habitat and reef extent or morphology' be better way to describe this and keep the terminology more consistent? Also as you then say 'morphological adjustments on line 85

Line 84: again there seem to be lots of different ways of describing what I deem to be the same thing (habitat type) but all mean something slightly different e.g. benthic composition, coral habitat type, coral communities, habitat type - it just needs tightening on the terminology

Line 85: I wouldn't use the word 'response' in relation to coral communities in this context - implies a current response whereas what you're predicting is change in relative area cover

Line 121: how was rugosity measured and modelled forward and backward? Perhaps a few more details on this would be good.

Line 122: key habitat types mentioned here for the first time – but no idea what these are until later on. I would put them in brackets here

Line 138: 10x10 m grid – this could be interpreted as a 10 by 10 m total grid size (which is not the case) or that each square on the grid is 10 by 10 m

Line 143: change ‘force’ to ‘forced’

Line 163: ‘annual reef accretion rates’ or $g(d)$ – does this all relate to paragraph 187 to 192? If so, perhaps this paragraph could be better placed?

Line 171 to 177: It seems that how L is calculated (with $L=0.25$ m year⁻¹) – is critical to then work out the rate of reef lateral expansion. But aside from saying that this number represents the age difference between consecutive reef cores, there is little information here on this. So I assume this is the average of all these calculations? Is this based on age differences at all points in time (so with depth)?

Line 188 – what’s a site? A grid square?

Line 195: ‘rates of SLR’ – you give details later on these – but first mention it here

Figure 1: is very confusing and apologies if I’m completely mis-interpreting it. In the text on line 218 it states that reef complex growth started between 1200 – 800 BP. But when I look at figure 1 it seems that PSS started 200 BP? Also in the figure caption it states that it’s the proportion of present day reef substrate whereas on the y axis it states present day coral?? I guess I don’t fully understand what that y axis represents.

Line 228 to 229 (starting – Reefs accreted slowly...’ Is this in reference to the 500 yrs BP – just needs clarification as written

GENERAL COMMENT:

I found that although the figures were well put together – some of the key points made in the text were not captured in figures and therefore could not be 1) observed and 2) verified. This applied to paragraphs from line 237 to 300. Some examples of the comments that were made that would be good to see visually include:

1. quantitative loss in productive shallow reef (line 282)
2. loss of shallow water habitat line 294 (very difficult to see depth differences in figure 5)
3. 25% of reef area dominated by turbinaria (line 296)
4. declines in coral cover from 46 to 13% - line 263

Instead in figures we largely see changes in relative habitat types and reef area. It would be good if the text better matched some of the key points that were coming across – particularly in figure 5. Maybe some sort of graph that combines coral cover, reef accretion and depth???

Line 246-247: goes back to one of my first points about changes in diversity (see previous comment)

General comment: you have used m² and km² but also hectares – be good to keep to the same type of units for consistency

Line 262: in reference to figure 4 – I can’t see the related comment coming through in this figure. Figure 5 and S5 don’t fully match – I think that the data for H4 and H5 have been mixed around either in the graph and table. Plus I don’t see the relevance of the table given that the data is in the figure

Figure 5 – hard to interpret differences in depth between scenarios

Figure 4 and 5 – the colour for the H4 line doesn’t match the legend

Discussion:

I found the discussion well written – my only suggestion is that these future predictions may be over-estimating reef accretion given other future impacts from SST and OA – and how these might also impact rates of carbonate production. Perhaps a comment to that effect would be good?

Review form: Reviewer 2

Recommendation

Reject – article is not of sufficient interest (we will consider a transfer to another journal)

Scientific importance: Is the manuscript an original and important contribution to its field?

Good

General interest: Is the paper of sufficient general interest?

Marginal

Quality of the paper: Is the overall quality of the paper suitable?

Good

Is the length of the paper justified?

Yes

Should the paper be seen by a specialist statistical reviewer?

No

Do you have any concerns about statistical analyses in this paper? If so, please specify them explicitly in your report.

No

It is a condition of publication that authors make their supporting data, code and materials available - either as supplementary material or hosted in an external repository. Please rate, if applicable, the supporting data on the following criteria.

Is it accessible?

N/A

Is it clear?

N/A

Is it adequate?

N/A

Do you have any ethical concerns with this paper?

No

Comments to the Author

Using a detailed dataset of sediment cores and ecological data from an inshore reef from the central Great Barrier Reef, the authors present a well written paper exploring the capacity of turbid-zone reefs to adapt to rising sea-level. Turbid zone reefs are unique in that changes in SLR over the Holocene and resulting changes in water-depth above the reef surface strongly influences community structure of reef-building corals due to changes in environmental conditions (sedimentation, flow and light). The central question: “how will the spatial extent and habitat configurations of coral reefs, that influence these physical functional roles, change into the future as global sea levels rise?” is a genuinely interesting and important point.

The modelling approach adopted in the manuscript is eloquent and spatially realistic, but the results are often highly technical and difficult to follow. That reef geomorphological evolution will therefore drive natural perturbations in coral cover, species composition and reef structural complexity is an important question, but the discussion is strongly limited to local observations of Paluma shoals (changes in nearshore bathymetry, increases in “*Turbinaria* and *Porites*” corals),

and I'm left wondering how this will apply to other turbid-zone reefs on the GBR, or more broadly to other reef areas where depth is less of a constraint or driver of community structure. While the authors acknowledge that turbid-zone corals are more resistant to thermal bleaching, the accretion model is missing a response to SST changes (in both hindcast and forecast). If rapid accretion is driven by fast-growing taxa, how may increases in SST in the forecast (particularly under RCP 8.5) affect growth rates and survival under future scenarios (especially under a 500 year projection)?

Minor comments

In Figure 5, is the total reef-area (grey bars) correct? There seems to be an expansion of reef area under RCP scenarios in the northern reefs ("H7") compared to the static that isn't reflected in the total reef area.

Review form: Reviewer 3

Recommendation

Accept with minor revision (please list in comments)

Scientific importance: Is the manuscript an original and important contribution to its field?

Excellent

General interest: Is the paper of sufficient general interest?

Good

Quality of the paper: Is the overall quality of the paper suitable?

Excellent

Is the length of the paper justified?

Yes

Should the paper be seen by a specialist statistical reviewer?

Yes

Do you have any concerns about statistical analyses in this paper? If so, please specify them explicitly in your report.

No

It is a condition of publication that authors make their supporting data, code and materials available - either as supplementary material or hosted in an external repository. Please rate, if applicable, the supporting data on the following criteria.

Is it accessible?

Yes

Is it clear?

Yes

Is it adequate?

Yes

Do you have any ethical concerns with this paper?

No

Comments to the Author

This is a very well-written manuscript that provides an important perspective on the future evolution of nearshore, turbid reef ecosystems under scenarios of future sea-level rise. The combination of well-replicated paleoecological data with reef-habitat modeling provides a unique opportunity to use insights from reef histories to project reef futures. The results of this study are of particular importance given the fact that a number of recent studies have shown that turbid reefs may provide refugia from thermal stress. I just have a few, mostly minor, comments for the authors consideration and I suggest that the manuscript be accepted after minor revisions.

General comment: The ability of the authors to reconstruct/project changes in coral cover and rugosity with changing sea level is a unique aspect of this study and this data is something that I think a lot of researchers would be interested in. Because of that, it would be really useful if the authors could include a figure that shows how modeled coral cover and rugosity change through time and add some additional text related to these results to the Discussion.

Specific comments:

L74-75: It's not clear to me what this sentence means. The reefs are in a marginal environment and more turbid habitats do buffer reefs from bleaching, but I don't understand how marginal conditions are "attributed to" bleaching resistance. Consider rephrasing.

L103: Can you provide a depth range for the deeper, offshore reefs?

L104-106: Does tidal range also contribute to the high turbidity in this area?

L129-132: I'm wondering why accretion rate and community structure in the cores were compared with depth relative to LAT rather than paleo-water depth. Reefs grew between 2000 and 700 BP and although sea-level change over this period was likely minimal, paleodepth could have changed through time. I see now that L216-217 suggests static sea level over the period represented in the cores. This is important justification that should be added to the methods.

L141: "R" is already used as an abbreviation for rugosity. I would suggest using a different variable abbreviation for the presence/absence of reef.

L143: is each cell also assigned a single average depth based on the DEM?

L173: Uniform lateral expansion with depth makes sense, but adjacent cells do not necessarily have the same depth, correct? Can you clarify this.

L182-185: Does this mean that a reef cell cannot accrete vertically until it is completely filled? Please clarify.

L220: I'd add a citation to the original study here since the ages from the cores are not provided in this study.

L227-235: Please add references to the relevant supplementary tables and/or the original studies describing the core records.

L331: I would be more specific about the timing of the sea-level fall so that it's clear that sea level was stable during the time period considered in this study.

Fig. 4: should the secondary y-axis be "Total reef area" as in Fig. 5? "Total coral area" makes it sound like the figure is showing modeled coral cover, which it is not, correct?

Decision letter (RSPB-2020-0541.R0)

25-Apr-2020

Dear Dr Morgan:

Your manuscript has now been peer reviewed and the reviews have been assessed by an Associate Editor. The reviewers' comments (not including confidential comments to the Editor) and the comments from the Associate Editor are included at the end of this email for your

reference. As you will see, the reviewers and the Editors have raised some concerns with your manuscript and we would like to invite you to revise your manuscript to address them.

Research ethics:

Use of animals and field studies:

Please submit a copy of your revised paper within three weeks. If we do not hear from you within this time your manuscript will be rejected. If you are unable to meet this deadline please let us know as soon as possible, as we may be able to grant a short extension.

Best wishes,
Dr Daniel Costa
mailto:proceedingsb@royalsociety.org

Associate Editor
Board Member: 1
Comments to Author:

We have now received three expert reviews of your manuscript 'Projections of coral cover and habitat loss on turbid reefs submerged by rising sea levels'. All three reviewers regard this as an interesting and useful study. Although one reviewer suggests the manuscript may be better suited to a more specialised journal, the two other reviewers consider only minor revisions are necessary. These primarily relate to greater clarity in the description of model development and methods used and a closer alignment of the Figures and text (which may require some modifications to the Figures). My own reading of the manuscript generally supports these main concerns. Reviewer 1, in particular, has provided detailed comments which would substantially improve the overall attraction of the manuscript to a diverse readership. From my perspective, it would also be useful to add some comment on how relevant the findings for this particular study site are for other turbid reefs (e.g. do we have an idea of what proportion of reefs on the Great Barrier Reef (or globally) fall into this turbid-zone classification?).

Reviewer(s)' Comments to Author:

Referee: 1

Comments to the Author(s)

The manuscript entitled 'Projections of coral cover and habitat loss on turbid reefs submerged by rising sea level' provides novel and useful insights into how these marginal reef types will respond to SLR in regards to their accretionary potential and habitat structure. The authors highlight the importance of understanding how these reefs will respond to SLR and the rates of change given that these reefs have been linked to increased resilience to sea surface temperature

and may therefore be potential coral refugia hotspots. The manuscript is well written and the model development is (mostly) well explained. The modelling relies on one of the largest collection of reef cores available globally for a relatively small reef complex and together with detailed habitat maps of the present day reef – is based on a large and robust data set. As such, I recommend the paper for publication with minor changes (see below). Most of the required changes relate either to more clarification around terminology and methods, and the potential for new or replacement figures that better match the text descriptions.

Line 40: ‘reductions in coral cover and diversity’ I found it hard to actually ‘see’ the decline in coral cover in the manuscript. This was described in the text but not visually represented in an accessible manner – only relative changes in the habitat types. Further the term diversity here is misleading for two reasons. The authors are referring to habitat diversity rather than coral diversity (so could be mis-interpreted) but also in the figures – you can see that there is a change in the relative contribution of the 7 habitat types but all 7 are present most of the time – so no major changes in diversity at this level.

Line 42 to 44: I agree with this line – which is largely based on the finding that rates of SLR under RCP8.5 are above the fastest rates of accretion seen on these turbid reefs (6.9 mm/year). But this is quite fast generally right?? So could this statement be taken beyond turbid reefs or is this more significant on turbid reefs given the turbidity driven reduced light levels below 4 m depth?

Line 74: ‘These marginal conditions...’ This is a confusing sentence and sounds like the environment depend on corals. I know what you mean – but needs re-writing.

Line 82: ‘reef configuration’ – would ‘habitat and reef extent or morphology’ be better way to describe this and keep the terminology more consistent? Also as you then say ‘morphological adjustments on line 85

Line 84: again there seem to be lots of different ways of describing what I deem to be the same thing (habitat type) but all mean something slightly different e.g. benthic composition, coral habitat type, coral communities, habitat type – it just needs tightening on the terminology

Line 85: I wouldn’t use the word ‘response’ in relation to coral communities in this context – implies a current response whereas what you’re predicting is change in relative area cover

Line 121: how was rugosity measured and modelled forward and backward? Perhaps a few more details on this would be good.

Line 122: key habitat types mentioned here for the first time – but no idea what these are until later on. I would put them in brackets here

Line 138: 10x10 m grid – this could be interpreted as a 10 by 10 m total grid size (which is not the case) or that each square on the grid is 10 by 10 m

Line 143: change ‘force’ to ‘forced’

Line 163: ‘annual reef accretion rates’ or g(d) – does this all relate to paragraph 187 to 192? If so, perhaps this paragraph could be better placed?

Line 171 to 177: It seems that how L is calculated (with $L=0.25$ m year⁻¹) – is critical to then work out the rate of reef lateral expansion. But aside from saying that this number represents the age difference between consecutive reef cores, there is little information here on this. So I assume this is the average of all these calculations? Is this based on age differences at all points in time (so with depth)?

Line 188 – what’s a site? A grid square?

Line 195: ‘rates of SLR’ – you give details later on these – but first mention it here

Figure 1: is very confusing and apologies if I’m completely mis-interpreting it. In the text on line 218 it states that reef complex growth started between 1200 – 800 BP. But when I look at figure 1 it seems that PSS started 200 BP? Also in the figure caption it states that it’s the proportion of present day reef substrate whereas on the y axis it states present day coral?? I guess I don’t fully understand what that y axis represents.

Line 228 to 229 (starting – Reefs accreted slowly...’ Is this in reference to the 500 yrs BP – just needs clarification as written

GENERAL COMMENT:

I found that although the figures were well put together – some of the key points made in the text where not captured in figures and therefore could not be 1) observed and 2) verified. This

applied to paragraphs from line 237 to 300. Some examples of the comments that were made that would be good to see visually include:

1. quantitative loss in productive shallow reef (line 282)
2. loss of shallow water habitat line 294 (very difficult to see depth differences in figure 5)
3. 25% of reef area dominated by turbinaria (line 296)
4. declines in coral cover from 46 to 13% - line 263

Instead in figures we largely see changes in relative habitat types and reef area. It would be good if the text better matched some of the key points that were coming across – particularly in figure 5. Maybe some sort of graph that combines coral cover, reef accretion and depth???

Line 246-247: goes back to one of my first points about changes in diversity (see previous comment)

General comment: you have used m² and km² but also hectares – be good to keep to the same type of units for consistency

Line 262: in reference to figure 4 – I can't see the related comment coming through in this figure. Figure 5 and S5 don't fully match – I think that the data for H4 and H5 have been mixed around either in the graph and table. Plus I don't see the relevance of the table given that the data is in the figure

Figure 5 – hard to interpret differences in depth between scenarios

Figure 4 and 5 – the colour for the H4 line doesn't match the legend

Discussion:

I found the discussion well written – my only suggestion is that these future predictions may be over-estimating reef accretion given other future impacts from SST and OA – and how these might also impact rates of carbonate production. Perhaps a comment to that effect would be good?

Referee: 2

Comments to the Author(s)

Using a detailed dataset of sediment cores and ecological data from an inshore reef from the central Great Barrier Reef, the authors present a well written paper exploring the capacity of turbid-zone reefs to adapt to rising sea-level. Turbid zone reefs are unique in that changes in SLR over the Holocene and resulting changes in water-depth above the reef surface strongly influences community structure of reef-building corals due to changes in environmental conditions (sedimentation, flow and light). The central question: “how will the spatial extent and habitat configurations of coral reefs, that influence these physical functional roles, change into the future as global sea levels rise?” is a genuinely interesting and important point.

The modelling approach adopted in the manuscript is eloquent and spatially realistic, but the results are often highly technical and difficult to follow. That reef geomorphological evolution will therefore drive natural perturbations in coral cover, species composition and reef structural complexity is an important question, but the discussion is strongly limited to local observations of Paluma shoals (changes in nearshore bathymetry, increases in “Turbinaria and Porites” corals), and I'm left wondering how this will apply to other turbid-zone reefs on the GBR, or more broadly to other reef areas where depth is less of a constraint or driver of community structure. While the authors acknowledge that turbid-zone corals are more resistant to thermal bleaching, the accretion model is missing a response to SST changes (in both hindcast and forecast). If rapid accretion is driven by fast-growing taxa, how may increases in SST in the forecast (particularly under RCP 8.5) affect growth rates and survival under future scenarios (especially under a 500 year projection)?

Minor comments

In Figure 5, is the total reef-area (grey bars) correct? There seems to be an expansion of reef area under RCP scenarios in the northern reefs ("H7") compared to the static that isn't reflected in the total reef area.

Referee: 3

Comments to the Author(s)

This is a very well-written manuscript that provides an important perspective on the future evolution of nearshore, turbid reef ecosystems under scenarios of future sea-level rise. The combination of well-replicated paleoecological data with reef-habitat modeling provides a unique opportunity to use insights from reef histories to project reef futures. The results of this study are of particular importance given the fact that a number of recent studies have shown that turbid reefs may provide refugia from thermal stress. I just have a few, mostly minor, comments for the authors consideration and I suggest that the manuscript be accepted after minor revisions.

General comment: The ability of the authors to reconstruct/project changes in coral cover and rugosity with changing sea level is a unique aspect of this study and this data is something that I think a lot of researchers would be interested in. Because of that, it would be really useful if the authors could include a figure that shows how modeled coral cover and rugosity change through time and add some additional text related to these results to the Discussion.

Specific comments:

L74-75: It's not clear to me what this sentence means. The reefs are in a marginal environment and more turbid habitats do buffer reefs from bleaching, but I don't understand how marginal conditions are "attributed to" bleaching resistance. Consider rephrasing.

L103: Can you provide a depth range for the deeper, offshore reefs?

L104-106: Does tidal range also contribute to the high turbidity in this area?

L129-132: I'm wondering why accretion rate and community structure in the cores were compared with depth relative to LAT rather than paleo-water depth. Reefs grew between 2000 and 700 BP and although sea-level change over this period was likely minimal, paleodepth could have changed through time. I see now that L216-217 suggests static sea level over the period represented in the cores. This is important justification that should be added to the methods.

L141: "R" is already used as an abbreviation for rugosity. I would suggest using a different variable abbreviation for the presence/absence of reef.

L143: is each cell also assigned a single average depth based on the DEM?

L173: Uniform lateral expansion with depth makes sense, but adjacent cells do not necessarily have the same depth, correct? Can you clarify this.

L182-185: Does this mean that a reef cell cannot accrete vertically until it is completely filled? Please clarify.

L220: I'd add a citation to the original study here since the ages from the cores are not provided in this study.

L227-235: Please add references to the relevant supplementary tables and/or the original studies describing the core records.

L331: I would be more specific about the timing of the sea-level fall so that it's clear that sea level was stable during the time period considered in this study.

Fig. 4: should the secondary y-axis be "Total reef area" as in Fig. 5? "Total coral area" makes it sound like the figure is showing modeled coral cover, which it is not, correct?

Author's Response to Decision Letter for (RSPB-2020-0541.R0)

See Appendix A.

Decision letter (RSPB-2020-0541.R1)

24-May-2020

Dear Dr Morgan

I am pleased to inform you that your manuscript entitled "Projections of coral cover and habitat change on turbid reefs under future sea level rise" has been accepted for publication in Proceedings B.

Open Access

Paper charges

Sincerely,

Dr Daniel Costa

Associate Editor:

Board Member

Comments to Author:

Thank you for your revised manuscript and detailed responses to the Reviewers' comments.

Appendix A

Board Member: We have now received three expert reviews of your manuscript 'Projections of coral cover and habitat loss on turbid reefs submerged by rising sea levels'. All three reviewers regard this as an interesting and useful study. Although one reviewer suggests the manuscript may be better suited to a more specialised journal, the two other reviewers consider only minor revisions are necessary. These primarily relate to greater clarity in the description of model development and methods used and a closer alignment of the Figures and text (which may require some modifications to the Figures). My own reading of the manuscript generally supports these main concerns. Reviewer 1, in particular, has provided detailed comments which would substantially improve the overall attraction of the manuscript to a diverse readership.

We would like to thank the Editorial Board Member and the three reviewers for their constructive comments that have improved our manuscript. We are very pleased that the reviewers found the study of interest and that they recognise the importance of utilising cross-disciplinary datasets to better understand one of the lesser studied impacts of climate change on coral reefs – future sea level rise (SLR).

We have carefully considered each of the reviewer's comments, particularly R1, and have made a number of revisions within the manuscript to address these. Specifically, we have addressed the following:

- Provided additional detail where requested and reworked sentences that were unclear
- Used consistent terminology throughout the manuscript when discussing habitat extent
- Made revisions to the Results section so the text more clearly corresponds to figures
- Revised Figure 5 by creating histograms to show changes in habitat type, reef depth, coral cover and reef rugosity under different projections of SLR
- Contextualised our findings by discussing the GBR/global distribution of turbid reefs

From my perspective, it would also be useful to add some comment on how relevant the findings for this particular study site are for other turbid reefs (e.g. do we have an idea of what proportion of reefs on the Great Barrier Reef (or globally) fall into this turbid-zone classification?).

We agree. Turbid reefs are relatively understudied environments and their distribution is often difficult to quantify because of poor water clarity. However, a recent paper has used MODIS-aqua satellite data to classify global turbid reefs (Sully & van Woerik (2020) *Global Change Biology*). Based on their findings, we have added a paragraph to the Discussion to contextualise our results in terms of the wider GBR and global occurrence of turbid coral reefs.

"Although our study focuses on the reefs that form the Paluma Shoals reef complex (because of the extensive and detailed ecological and geological datasets available from these sites) other nearby reefs on the GBR show similar spatial distributions in habitat type and coral community structure [31]. Further, a recent global analysis using Modis-Aqua satellite imagery has suggested that approximately 12% of the world's coral reefs exist within a "moderating turbidity" range, where turbidity is sufficient to mediate coral bleaching because of increased light attenuation [40]. Of the reefs located within the central and northern GBR ecoregion, 6.43% (562 km²) were considered turbid [40]. This highlights the wider implications of our findings beyond PSRC as we expect to see similar shifts in habitat extent and coral cover within these other coastal areas as sea levels rise. At an individual reef-scale, the timing and magnitude of habitat reconfiguration will be determined by three interacting factors: (1) the rate of SLR; (2) the rate of vertical reef accretion by coral communities; and (3) the local turbidity regime. Reefs that inhabit more turbid water will respond quickly because coral depth ranges are likely to be very vertically compressed. Conversely, similar increases in relative water depth on clear-water reefs, where depth is less of a driver of community structure, could have less immediate ecological impacts."

Referee 1: The manuscript entitled 'Projections of coral cover and habitat loss on turbid reefs submerged by rising sea level' provides novel and useful insights into how these marginal reef types will respond to SLR in regards to their accretionary potential and habitat structure. The authors highlight the importance of understanding how these reefs will respond to SLR and the rates of change given that these reefs have been linked to increased resilience to sea surface temperature

and may therefore be potential coral refugia hotspots. The manuscript is well written and the model development is (mostly) well explained. The modelling relies on one of the largest collection of reef cores available globally for a relatively small reef complex and together with detailed habitat maps of the present day reef – is based on a large and robust data set. As such, I recommend the paper for publication with minor changes (see below). Most of the required changes relate either to more clarification around terminology and methods, and the potential for new or replacement figures that better match the text descriptions.

We thank Reviewer 1 for their detailed comments that have improved the manuscript. We are pleased that R1 recognises the significance of the compiled dataset from which our analyses are derived. We have made a number of revisions based on the R1's comments, including using consistent terminology throughout the manuscript, a better correlation results text and figures, and reworked figures to more explicitly show projected change (Fig. 5). Responses to each of R1's specific comments are addressed below.

Line 40: 'reductions in coral cover and diversity' I found it hard to actually 'see' the decline in coral cover in the manuscript. This was described in the text but not visually represented in an accessible manner – only relative changes in the habitat types. Further the term diversity here is misleading for two reasons. The authors are referring to habitat diversity rather than coral diversity (so could be misinterpreted) but also in the figures – you can see that there is a change in the relative contribution of the 7 habitat types but all 7 are present most of the time – so no major changes in diversity at this level.

We have revised the results section and provided new additional data plots in Figure 5 to more explicitly show changes (habitat, depth, coral cover, rugosity) we describe in Line 40.

In this instance, we were referring to coral diversity (not habitat diversity), specifically coral generic diversity. Based on our data from living coral communities, and the classification of this extensive spatial dataset into different habitat types, we have a very good understanding of the dominant coral taxa within each habitat. Although our model cannot project specific diversity values, our habitat classifications assume that changes in coral composition occur (driven by shifts in habitat). For example, if a habitat transitions from Montipora and Acropora framework (which comprise more diverse coral assemblages) to Turbinaria carpets (monospecific coral communities) this would indicate a projected reduction in generic diversity. To avoid confusion, we have reworded this sentence.

"Model outputs show that modest increases in relative water depth above reefs (RCP4.5) over the next 100 years will increase the extent of habitats with low coral cover and generic diversity."

We agree that our findings typically show changes in relative habitat abundance rather than habitat diversity (which implies a loss of gain of habitats). We have removed references to "habitat diversity" throughout. Instead "habitat extent" is used.

Line 42 to 44: I agree with this line – which is largely based on the finding that rates of SLR under RCP8.5 are above the fastest rates of accretion seen on these turbid reefs (6.9 mm/year). But this is quite fast generally right?? So could this statement be taken beyond turbid reefs or is this more significant on turbid reefs given the turbidity driven reduced light levels below 4 m depth?

Here we are referring to turbid reefs because our model is parameterised using data specific to these environments, and which likely differ from other reefs (e.g. clear-water systems). Although we agree that SLR will influence all coral reefs, especially as rates of vertical reef accretion on clear-water reefs are often much lower, we feel the impacts of SLR will be more rapid on severe within turbid settings because of the importance of water depth and light attenuation in these settings.

"Our findings suggest adverse future trajectories associated with high emission climate scenarios which could threaten turbid reefs globally and their capacity to act as coral refugia from climate change."

Line 74: 'These marginal conditions...' This is a confusing sentence and sounds like the environment depend on corals. I know what you mean – but needs re-writing.

We agree, this sentence has been revised.

"The relationship between habitat type and the relative water level above reefs results from the rapidly changing light, wave and sedimentary conditions within coastal areas of the GBR [17] that may also provide turbid corals with a greater resistance to thermal bleaching [18]."

Line 82: 'reef configuration' – would 'habitat and reef extent or morphology' be better way to describe this and keep the terminology more consistent? Also as you then say 'morphological adjustments on line 85

We have removed this sentence due to general editing of the text. We have incorporated this comment into the next sentence.

"Model outputs are used to quantify changes in habitat extent following adjustments of the underlying reef morphology as it grows vertically."

Line 84: again there seem to be lots of different ways of describing what I deem to be the same thing (habitat type) but all mean something slightly different e.g. benthic composition, coral habitat type, coral communities, habitat type – it just needs tightening on the terminology

We have reworded the text to keep more consistent throughout. However, in some stances we do require the use of more detail descriptive terms to better portray our findings.

"Model outputs are used to quantify changes in habitat extent following adjustments of the underlying reef morphology as it grows vertically."

Line 85: I wouldn't use the word 'response' in relation to coral communities in this context – implies a current response whereas what you're predicting is change in relative area cover

We have removed "response" and reworded the sentence based on these comments.

"Model outputs are used to quantify changes in habitat extent following adjustments of the underlying reef morphology as it grows vertically."

Line 121: how was rugosity measured and modelled forward and backward? Perhaps a few more details on this would be good.

The rugosity of coral reef framework was evaluated qualitatively for each individual still frame using a modified version of the scale developed by Polunin & Roberts (1993) described within Morgan et al., 2016 (referenced at the end of this sentence). Median rugosity values are associated with different habitat types. Therefore, as habitat changes through time we can infer changes in framework rugosity values.

This rugosity scale ranges from 1-5, as per below:

- 1 - Flat and featureless, typically soft-sediments and/muds. No three-dimensionality.
- 2 - Low rugosity coarse gravel/rubble with isolated dead or live coral framework
- 3 - Moderate rugosity, high coral cover comprising typically low-profile colonies (e.g. encrusting or tabular).
- 4 - High rugosity, mixed structurally-complex coral framework (branching, tabular and foliose).
- 5 - Very high rugosity (associated with high coral cover), high three-dimensionality with many crevices and overhangs.

Polunin, N. V. C., & Roberts, C. M. (1993). Greater biomass and value of target coral-reef fishes in two small Caribbean marine reserves. *Marine Ecology-Progress Series*, 100, 167-167.

We have added additional detail in the text. However, given the length of methods section that requires specific details on the modelling procedure we have kept this brief.

“Each frame was depth-calibrated using the DEM and a digital 9-point grid overlay was added to frames to calculate benthic community composition (%) and make a qualitative estimate (scaled 1-5) of reef rugosity (R) [15].”

Line 122: key habitat types mentioned here for the first time – but no idea what these are until later on. I would put them in brackets here

We agree. A list of habitat types has been included as suggested.

“The depth ranges of dominant coral genera and key habitat types were then established (H1: *Goniastrea* reef flat (> LAT); H2: rubble and encrusting coral (0-0.5 m); H3: *Montipora* and *Acropora* framework (0.5-1.5 m); H4: massive *Porites* and sand (1.5-2 m); H5: *Turbinaria* carpets (2-3 m); H6: rubble and sediment-tolerant coral (3-4 m); H7: sand/mud (> 4 m)).”

Line 138: 10x10 m grid – this could be interpreted as a 10 by 10 m total grid size (which is not the case) or that each square on the grid is 10 by 10 m

We have clarified that the total DEM size is 15 km² comprising grid cells sized 10 x 10 m.

“The DEM of seafloor and reef morphology (total area: 15 km²; 10 x 10 m grid cells) was used to model past and future reef growth (+/- 500 years) and to quantify habitat extent.”

Line 143: change ‘force’ to ‘forced’

We have changed this word.

Line 163: ‘annual reef accretion rates’ or $g(d)$ – does this all relate to paragraph 187 to 192? If so, perhaps this paragraph could be better placed?

We agree with the reviewer. We have moved this section to earlier in the text to follow methods for calculating vertical reef accretion.

Line 171 to 177: It seems that how L is calculated (with $L=0.25$ m year⁻¹) – is critical to then work out the rate of reef lateral expansion. But aside from saying that this number represents the age difference between consecutive reef cores, there is little information here on this. So I assume this is the average of all these calculations? Is this based on age differences at all points in time (so with depth)?

We provide details on how lateral expansion was calculated and a reference to a study from which the approached was utilised.

Palmer SE, Perry CT, Smithers SG, Gulliver P. 2010 Internal structure and accretionary history of a nearshore, turbid-zone coral reef: Paluma Shoals, central Great Barrier Reef, Australia. *Mar. Geol.* 276, 14–29. (doi:10.1016/j.margeo.2010.07.002)

In response to the reviewer’s question – yes, this was derived from the average of these calculations. An average was used because of the associated variability in ages/depths and distance between cores on transects. We have added text to clarify how this was done.

“To model lateral reef growth, we use a lateral expansion rate ($L=0.25$ m y⁻¹) calculated from the average age difference between consecutive reef cores at the same horizontal distance on a transect, and the horizontal distance between these cores [23].”

Line 188 – whats a site? A grid square?

A site refers to a grid square. We revised the manuscript to be more consistent in our terminology and have changed all references to sites/grid squares to “grid cells”.

“First, we assume the amount of vertical accretion at every grid cell is independent of every other.”

Line 195: 'rates of SLR' – you give details later on these – but first mention it here

We thank the reviewer for pointing this out. We have moved details on the rates of sea level rise used in future model projections to this point in the text.

“We then incorporated rates of sea level rise (r) projected by IPCC Representative Concentration Pathways (RCP) 4.5 (5.5 mm y^{-1}) and 8.5 (7.5 mm y^{-1}) climate change scenarios [21].”

Figure 1: is very confusing and apologies if I'm completely mis-interpreting it. In the text on line 218 it states that reef complex growth started between 1200 – 800 BP. But when I look at figure 1 it seems that PSS started 200 BP? Also in the figure caption it states that it's the proportion of present day reef substrate whereas on the y axis it states present day coral?? I guess I don't fully understand what that y axis represents.

Figure 1 shows the amount of reef remaining on the seafloor for each identified structure (coloured reefs correspond to graph curves). We use our predictive model to shrink the reefs backwards (from 100% at present day), using rates of vertical accretion and lateral expansion, to their initiation point (defined as when there is no remaining reef on the seafloor – 0%). We do this to see if there is suitable agreement between model and core timing.

We agree with the reviewer that there is discrepancy between the timing reported in the text and the figure, and thank the reviewer for bringing this to our attention. This occurred because our analysis incorporated all reef structures identified from bathymetry and ecological surveys, including small and deeper incipient reefs (PSN & blue, purple, grey structures) which produced the young modelled initiation ages (200-600 yrs BP). However, we do not have core data from these sites to confirm their chronology. The ages ranges reported in the text (1200-800 BP) are from the larger main reef complexes (e.g. PS, OPS, OPSA, OPSB, OPSC, OPSD) where core and age data is available, and which do show a very close match to our modelled data. We have clarified this in the text.

“By shrinking the reefs backwards, the model indicates that the main reef complexes started growing ~1200-700 years before present day (yrs BP) (Fig. 1). The modelled timings of reef initiation align with the reef-building phases discerned from the basal ages in cores at these sites (~1400-700 yrs BP) [16]. There are several smaller reef structures evident in our bathymetry and ecological datasets. We do not have core data from these to test the model against, but assuming they have had similar growth histories to the more established main reefs, our model suggests these would have established somewhere between 600 and 200 yrs BP (Fig. 1).

In addition, we have also modified the Y-axis to “Remaining reef area (%)”, updated the figure caption, and arranged the ordering of the curves to make the main reef complexes more prominent.

“Figure 1. Percent (%) of remaining reef area as reef morphology is modelled backwards from present day to reef initiation (i.e. start-up). Model simulations assume no sea level change and constant (but depth-variable) reef accretion rates. Lines represent the min/max estimated change from a 95% confidence window using 100 bootstrap samples calculated for different seafloor extrapolations (see Materials and Methods). Colours indicate the reefs that form the Paluma Shoals reef complex (Paluma Shoals (PS), Offshore Paluma Shoals (OPS) A-D). “

Line 228 to 229 (starting – Reefs accreted slowly...’ Is this in reference to the 500 yrs BP – just needs clarification as written

Yes, it is. We have rewritten this sentence to improve clarity.

“Core records suggest that by 500 yrs BP reefs comprising sediment-tolerant coral genera (e.g. Lobophyllia, Goniopora, Galaxea) had initiated early framework accumulation that accreted slowly (2.4 mm y^{-1}) for several hundred years [16].”

GENERAL COMMENT:

I found that although the figures were well put together – some of the key points made in the text where not captured in figures and therefore could not be 1) observed and 2) verified. This applied to

paragraphs from line 237 to 300. Some examples of the comments that were made that would be good to see visually include:

1. quantitative loss in productive shallow reef (line 282)
2. loss of shallow water habitat line 294 (very difficult to see depth differences in figure 5)
3. 25% of reef area dominated by turbinaria (line 296)
4. declines in coral cover from 46 to 13% - line 263

Instead in figures we largely see changes in relative habitat types and reef area. It would be good if the text better matched some of the key points that were coming across – particularly in figure 5. Maybe some sort of graph that combines coral cover, reef accretion and depth???

We have made a number of revisions to the Results text (line 240-299) to more explicitly capture points 1-4. We feel that there is now a much closer alignment between the text and the figures. In particular, we have focused on changes in habitat and what this means for associated coral cover and rugosity based on our datasets of living coral communities.

In addition, we provide a replacement Figure 5 that quantifies spatial changes in habitat extent, reef depth, coral cover and reef rugosity under different SLR projections. We chose to separate this data for PS (shore-attached) and OPS (shore-detached) reefs because the response of reefs to SLR differs between them. These patterns were previously lost when discussing PSRC as a whole reef system. We hope this improves clarity and readability of the results.

Line 246-247: goes back to one of my first points about changes in diversity (see previous comment)

We use the term “habitat diversity” to refer to the increasing representation of a wider range of habitat types as reef morphology develops. However, we agree this does not infer a change in diversity as such. This sentence has been removed.

General comment: you have used m² and km² but also hectares – be good to keep to the same type of units for consistency

We have used different units when reporting values because of the different scales involved when discussing grid squares on the model (m²) versus total modelled area or habitat extent across the entire reef complex (km²/ha). We have tried to avoid unnecessarily large or small values when presenting analyses. The units are commonly used and conversion between them is simple. However, we do agree these units should be more consistent and we have changed several throughout.

Line 262: in reference to figure 4 – I can't see the related comment coming through in this figure.

This description is in reference to the increase in relative cover of Habitat 3 (light blue; Montipora and Acropora habitat) from +100 years to +300 years. This habitat is associated with high vertical accretion rates because of the fast-growing corals and resultant carbonate production. This period of rapid-building coincides with increasing relative cover of Habitat 1 (red; Goniastrea reef flat) from +200 years. As the proportion of reef flat increases by +400 years, we see a decline in Habitat 4 as conditions (depth) are no longer suitable for rugose coral framework.

We have reworded this paragraph based on this comment to better describe the observed patterns in Figure 4.

“The model shows declines in Turbinaria carpet habitat (H5: 2-3 m), and expansion of shallow-water Montipora and Acropora framework habitat (H3; 0.5 -1.5 m depth) to 32% of total reef area by +300 years (Fig. 4). Coral communities associated with the Montipora and Acropora framework habitat are characterized by high coral cover ($47 \pm 19\%$; median $R = 4$) and rapid vertical accretion (3.4 mm y⁻¹) (Supplementary Table S2). At these rates of reef growth, reef flat habitat (H1: >LAT) will form across 23% of reef surfaces (Figs. 3 & 4) between +300-500 years. Shifts in coral communities associated with these changes in habitat type suggest that coral cover on upper reef surfaces will decline from

47% at present day to 13% by +500 years, and transitions in benthic cover to exposure-tolerant massive corals (*Goniastrea*), encrusting corals (*Montipora*, *Galaxea*), rubble ($74 \pm 12.3\%$), and macroalgae ($2.5 \pm 3.9\%$) will occur (Supplementary Table S2).”

Figure 5 and S5 don't fully match – I think that the data for H4 and H5 have been mixed around either in the graph and table. Plus I don't see the relevance of the table given that the data is in the figure

We thank the reviewer for the comment, and based on this we have removed Table S4 and S5 from the ESM as we agree that the data is represented in the figures. We have also incorporated % relative habitat cover for each SLR scenario on each spatial map (Fig. 5) for easier reference.

Figure 5 – hard to interpret differences in depth between scenarios

The bathymetry and depth contour show the changes in depth between each of the 3 SLR projections. However, we appreciated that given the size of the maps we are limited to, the change may be difficult to observe. Therefore, as a part of our revised Fig.5 we have quantified area of PS and OPS reefs in each depth contour to more explicitly show change in water levels above reefs (i.e. depth).

Figure 4 and 5 – the colour for the H4 line doesn't match the legend

We have recoloured all lines and legends keys using Illustrator eyedropper tool to ensure they are the same colour match.

Discussion:

I found the discussion well written – my only suggestion is that these future predictions may be over-estimating reef accretion given other future impacts from SST and OA – and how these might also impact rates of carbonate production. Perhaps a comment to that effect would be good?

We thank the reviewer for their comments. When calculating future predictions there is always uncertainty. Our forward model projections have not included the potential impacts of SST or OA. Firstly, long-term paleoecological evidence throughout the growth history of these reefs shows no evidence of past SST coral mortality (Johnson et al., 2017), nor do recent assessments during periods of elevated heat stress (e.g. 2016; Morgan et al., 2017). We believe this is a strength of the current dataset – where the continuous growth of coral communities over time (with very little disturbance) provides a solid basis to model future reef trajectories. Further, mass bleaching-related mortality could actually lead to increased vertical reef accretion rates as the living carbonate framework collapses and is incorporated into the reef structure.

Secondly, the impacts of OA on coral reef bioconstruction at a reef-scale, particularly within turbid settings where the sediment matrix contains high proportions of non-carbonate sediments, is poorly understood. There are very few studies (if any) that sufficiently show that OA will have a meaningful impact on vertical reef growth over long-term time-scales.

Based on the above points, we consider that the discussion of the impacts of SST or OA on future reef growth would be purely speculative. We prefer to keep the focus of the discussion targeted to the impacts of future relative water depth changes on coral reefs.

Referee 2: Using a detailed dataset of sediment cores and ecological data from an inshore reef from the central Great Barrier Reef, the authors present a well written paper exploring the capacity of turbid-zone reefs to adapt to rising sea-level. Turbid zone reefs are unique in that changes in SLR over the Holocene and resulting changes in water-depth above the reef surface strongly influences community structure of reef-building corals due to changes in environmental conditions (sedimentation, flow and light). The central question: “how will the spatial extent and habitat configurations of coral reefs, that influence these physical functional roles, change into the future as global sea levels rise?” is a genuinely interesting and important point.

We thank the reviewer for constructive comments and are pleased that they find this study of interest and importance.

The modelling approach adopted in the manuscript is eloquent and spatially realistic, but the results are often highly technical and difficult to follow.

Based on this comment, and those from R1 (addressed above), we have revised the results section for readability. Attention has been given to make more explicit links between the text and the figures.

That reef geomorphological evolution will therefore drive natural perturbations in coral cover, species composition and reef structural complexity is an important question, but the discussion is strongly limited to local observations of Paluma shoals (changes in nearshore bathymetry, increases in “Turbinaria and Porites” corals), and I’m left wondering how this will apply to other turbid-zone reefs on the GBR, or more broadly to other reef areas where depth is less of a constraint or driver of community structure.

We have centred our discussion around PSRC because the collection of work we present here are (to our knowledge) is the most detailed and complete records of turbid corals reef geomorphology and (paleo)ecology that exist to date. Unfortunately, field data from other turbid reefs is limited, but where data is available, it shows similar patterns in coral composition and habitat distribution.

We have added a paragraph (line 368 – 382) to the Discussion to contextualise our results in terms of the wider GBR and global occurrence of turbid coral reefs. Please see our response to the Board Member.

While the authors acknowledge that turbid-zone corals are more resistant to thermal bleaching, the accretion model is missing a response to SST changes (in both hindcast and forecast). If rapid accretion is driven by fast-growing taxa, how may increases in SST in the forecast (particularly under RCP 8.5) affect growth rates and survival under future scenarios (especially under a 500 year projection)?

As stated above in response to R1, our historical core datasets, paleoecological analysis and of living coral surveys during a period of extreme temperature stress on the GBR (2016), that caused widespread mortality of nearby clear-water reefs, do not show evidence of SST impacts on coral communities and/or reef accretion. Based on the data available, we do not believe it is suitable to incorporate SST events into future model projections. Further, there are no clear insights into the magnitude and/or frequency of bleaching events on coral reefs within contemporary coastal areas, nor into the future +500 years. Although we agree with the reviewer that major SST disturbance will impact future reefs, we simply do not have enough reliable data to model this at this time.

Minor comments

In Figure 5, is the total reef-area (grey bars) correct? There seems to be an expansion of reef area under RCP scenarios in the northern reefs (“H7”) compared to the static that isn’t reflected in the total reef area.

Based on comments received by R1, we have modified Fig. 5 and removed these plots.

Referee 3: This is a very well-written manuscript that provides an important perspective on the future evolution of nearshore, turbid reef ecosystems under scenarios of future sea-level rise. The combination of well-replicated paleoecological data with reef-habitat modeling provides a unique opportunity to use insights from reef histories to project reef futures. The results of this study are of particular importance given the fact that a number of recent studies have shown that turbid reefs may provide refugia from thermal stress. I just have a few, mostly minor, comments for the authors consideration and I suggest that the manuscript be accepted after minor revisions.

We thank the reviewer for their review of our manuscript and for the positive comments relating to our work. Their comments have been helpful in improving the manuscript.

General comment: The ability of the authors to reconstruct/project changes in coral cover and rugosity with changing sea level is a unique aspect of this study and this data is something that I think a lot of researchers would be interested in. Because of that, it would be really useful if the authors could

include a figure that shows how modeled coral cover and rugosity change through time and add some additional text related to these results to the Discussion.

We have modified Figure 5 based on the comments of R1 to include new histograms to quantify changes in habitat type, reef depth, coral cover and reef rugosity into the future (+100 years) under different SLR projections. We feel that this more clearly shows how these key ecological metrics are influenced by changes in reef morphology and relative water depth.

Specific comments:

L74-75: It's not clear to me what this sentence means. The reefs are in a marginal environment and more turbid habitats do buffer reefs from bleaching, but I don't understand how marginal conditions are "attributed to" bleaching resistance. Consider rephrasing.

We agree, and have revised this sentence to clarify its meaning.

"The relationship between habitat type and the relative water level above reefs results from the rapidly changing light, wave and sedimentary conditions within coastal areas of the GBR [17] that may also provide turbid corals with a greater resistance to thermal bleaching [18]."

L103: Can you provide a depth range for the deeper, offshore reefs?

We have provided the depth range of offshore reefs.

"(2) shore-detached reefs (Offshore Paluma Shoals (OPS) A, B, C & D) which remain submerged in deeper water further offshore (3-6 m LAT)."

L104-106: Does tidal range also contribute to the high turbidity in this area?

Yes, it does. We have added more detail relating to tidal currents and the tidal range.

"PSRC experiences high turbidity (up to 385 mg l⁻¹) and low light conditions because of the wave and tidal resuspension (tidal range: 3.6 m) of fine seafloor material that was reworked onshore during the last post-glacial transgression [12]."

L129-132: I'm wondering why accretion rate and community structure in the cores were compared with depth relative to LAT rather than paleo-water depth. Reefs grew between 2000 and 700 BP and although sea-level change over this period was likely minimal, paleodepth could have changed through time. I see now that L216-217 suggests static sea level over the period represented in the cores. This is important justification that should be added to the methods.

Studies from the central GBR suggest that sea level has been relatively static over the time-scales of reef development (review by Lewis et al. 2013; Quat. Sci. Rev.). We agree with the reviewer that this is an important justification and should be moved to earlier in the text to a more prominent position in the methods (Line 162-163).

"We then extrapolate backwards (Eq. 2) assuming a static sea level as suggested by Late Holocene sea-level data for the region [25], or forwards (Eq. 3), by subtracting or adding a growth term $g(d(X,Y,T))$."

L141: "R" is already used as an abbreviation for rugosity. I would suggest using a different variable abbreviation for the presence/absence of reef.

We have change the variable 'R' to 'P' to avoid confusion with rugosity (Line 140).

L143: is each cell also assigned a single average depth based on the DEM?

Yes, the grid cell height is a uniform depth derived from bathymetric data kriging to construct the digital elevation model (DEM).

L173: Uniform lateral expansion with depth makes sense, but adjacent cells do not necessarily have the same depth, correct? Can you clarify this.

Correct, adjacent cells do not have the same depth for a cell to expand. For example, reef cells can expand laterally into deeper water. However, if this cell is below the photic zone for reef growth (> 4m LAT) it will not grow vertically. We have added more detail to the text to explain this.

“We assume uniform expansion with depth, and model lateral reef expansion as follows: $p = L/s$, where s is the side length of a grid cell, then p is the proportion by which the reef grid cell expands each year into a neighboring non-reef grid cell (independent of adjacent cell depth).”

L182-185: Does this mean that a reef cell cannot accrete vertically until it is completely filled? Please clarify.

Correct, the adjacent cell must be filled before it can begin to accrete vertically. This assumes that lateral progradation can occur gradually via downslope transport of rubble. But the reef requires sufficient substrate for corals to colonise and begin to build reefs. We have added more detail to explain this.

“When $R \geq 1$ the reef has covered the entire cell, and we stop the accumulation and allow the new reef cell to grow vertically at the designated depth-assigned accretion rate (vertical accretion only begins once the cell is filled), as well as contribute to reef expansion into neighboring cells.”

L220: I'd add a citation to the original study here since the ages from the cores are not provided in this study.

We have added a citation here (and throughout) to reference the original study where all cores are presented.

L227-235: Please add references to the relevant supplementary tables and/or the original studies describing the core records.

We have added the appropriate references to the original studies and to ESM figs/table.

L331: I would be more specific about the timing of the sea-level fall so that it's clear that sea level was stable during the time period considered in this study.

This is good point. We have added the reported approximate timing of SL fall to avoid confusion.”

“Sea level fall on the GBR (~1-1.3 m at ~2000 yrs BP) during the Late Holocene have previously caused the “turn-off” of productive coral growth within shallow-water habitats, and a temporary hiatus of vertical reef growth as benthic communities shifted to rubble and macroalgal assemblages [33–35].”

Fig. 4: should the secondary y-axis be “Total reef area” as in Fig. 5? “Total coral area” makes it sound like the figure is showing modeled coral cover, which it is not, correct?

We thank the reviewer for pointing this out. Yes, we are referring to reef area. We have updated this in Fig. 4 – “Total reef area”.